# Trans4D: Realistic Geometry-Aware Transition for Compositional Text-to-4D Synthesis

## Abstract

Recent advances in diffusion models have demonstrated exceptional capabilities in image and video generation, further improving the effectiveness of 4D synthesis. Existing 4D generation methods can generate high-quality 4D objects or scenes based on user-friendly conditions, benefiting the gaming and video industries. However, these methods struggle to synthesize significant object deformation of complex 4D transitions and interactions within scenes. To address this challenge, we propose Trans4D, a novel text-to-4D synthesis framework that enables realistic complex scene transitions. Specifically, we first use multi-modal large language models (MLLMs) to produce a physic-aware scene description for 4D scene initialization and effective transition timing planning. Then we propose a geometry-aware 4D transition network to realize a complex scene-level 4D transition based on the plan, which involves expressive geometrical object deformation. Extensive experiments demonstrate that Trans4D consistently outperforms existing state-of-the-art methods in generating 4D scenes with accurate and high-quality transitions, validating its effectiveness.

## 1 Introduction

Recent diffusion model (DM) advances have revolutionized video and 3D synthesis. By harnessing the generative capability of DM, video generation methods (Liu et al., 2024b; Bao et al., 2024) have achieved high-quality video production that meets commercial standards. DreamFusion (Poole et al., 2023) introduced Score Distillation Sampling (SDS) to guide NeRF model optimization, marking a significant breakthrough in high-fidelity 3D generation.

Building on these remarkable breakthroughs, 4D generation methods have demonstrated impressive performance. These methods can be broadly categorized into three types: text-to-4D (Singer et al., 2023; Bahmani et al., 2024b; Zheng et al., 2024; Ling et al., 2024), single-image-to-4D (Zhao et al., 2023; Zheng et al., 2024), and monocular-video-to-4D (Ren et al., 2023; Jiang et al., 2024; Yin et al., 2023; Zeng et al., 2024; Zhang et al., 2024b; Wang et al., 2024a). Text-to-4D and Image-to-4D methods (Yu et al., 2024; Bahmani et al., 2024b; Zheng et al., 2024) combine video and multi-view generation models with SDS to synthesize 4D objects, though the motion remains limited due to current constraints in video generation models. Monocular-video-to-4D methods (Jiang et al., 2024; Wang et al., 2024a) utilize prior dynamics from video conditions to achieve high-quality 4D object synthesis with large-scale and natural motion, constrained by the requirement for videos with clear foreground subjects that are difficult to obtain. However, these methods primarily address local deformations of individual objects and fall short of generating complex 4D scenes that involve global interactions between multiple objects.

Rather than merely focusing on 4D object generation, text-to-4D methods like Comp4D (Xu et al., 2024) and monocular-video-to-4D methods such as Dreamscene4D (Chu et al., 2024) have achieved 4D scene generation. These methods still use deformation networks to adjust local coordinates and simulate movements of objects within 4D scenes, similar to 4D object generation methods. However, deformation networks are limited in handling significant object deformation in the 4D scene, which complicates the generation of 4D transitions with complex interactions, such as a missile transforming into an exploded cloud or a magician conjuring a dancer.

To address these challenges, we propose a text-to-4D method Trans4D, which leverages multimodal large language models (MLLMs) for geometry-aware 4D scene planning, and introduces a Transition

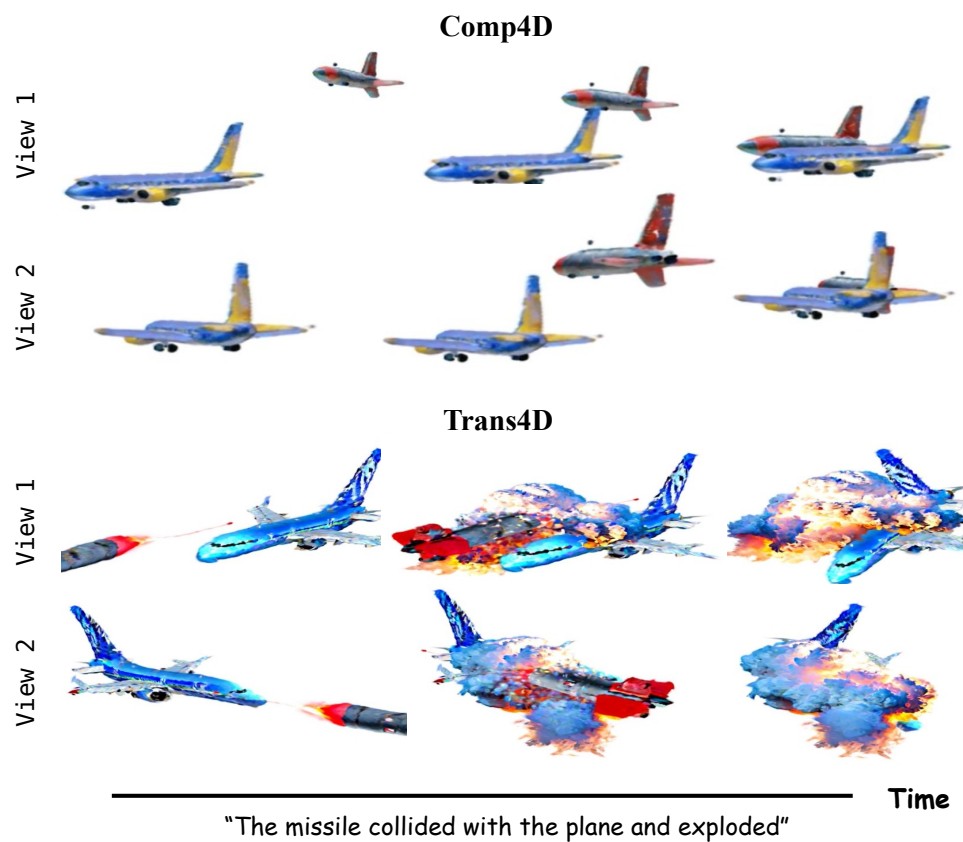

Figure 1: Comparing our TRANS4D with Comp4D (Xu et al., 2024) in 4D scene transition generation.

Network to simulate significant objects deformation within the generated 4D scenes. Unlike existing MLLMs that primarily describe or recognize input conditions, or methods like Comp4D (Xu et al., 2024) that focus on basic object trajectory function, we propose Physics-aware 4D Transition Planning method that enables MLLMs to generate detailed physical 4D information, including initial positions, movement and rotation speeds, and transition times. This allows for more precise 4D scene initialization and transition management. The Transition Network further realizes the transition process by predicting whether each point in the 3DGS model should appear or disappear at a specific time $t$. This capability ensures great control over transitions, enabling large-scale object transformations to be handled naturally and seamlessly, such as a missile transforming into an exploded cloud. As demonstrated in Fig. 1, our method achieves more natural and coherent 4D scene synthesis with complex interactions than existing text-to-4D scene generation techniques.

The main contributions of TRANS4D can be summarized as:

- In this work, we introduce a text-to-4D generation method called TRANS4D, which enables complex 4D scene synthesis and facilitates geometry-aware 4D scene transitions. Even if the 4D scene contains complex interactions or significant deformation among multiple objects, our method can stably generate high-quality 4D scenes.

- We present a Physics-aware 4D Transition Planning method, which sequentially leverages MLLM to perform physics-aware prompt expansion and transition planning. This approach ensures effective and reasonable initialization for 4D scene generation.

- We propose a geometry-aware Transition Network that achieves natural and smooth geometry-aware transitions in 4D scenes.

- Comprehensive experiments demonstrate that our TRANS4D generates more realistic and high-quality complex 4D scenes than existing baseline methods.

## 2 BACKGROUND & PROBLEM STATEMENT

### 2.1 4D CONTENT GENERATION

Research on 4D content generation begins with reconstructing dynamic 3D representations based on multi-view videos. Existing 4D reconstruction models (Pumarola et al., 2021; Wu et al., 2024a; Huang et al., 2024) achieve realistic 4D generation by extending 3D models such as NeRF and 3DGS. However, obtaining multi-view videos for 4D synthesis is challenging. Recently, more researchers have focused on 4D generation using simpler conditions, and these methods can be broadly divided into three categories: text-to-4D, image-to-4D, and monocular-video-to-4D. The text-to-4D (Singer et al., 2023; Bahmani et al., 2024b; Ling et al., 2024; Yu et al., 2024) and image-to-4D (Zhao et al., 2023; Zheng et al., 2024) methods are the first to be explored by researchers, typically extending 3D objects into 4D objects using SDS loss based on pretrained video DM. However, due to the limitations of SDS loss based on video DM, the dynamics of these 4D objects often seem unrealistic. Subsequently, some methods (Yin et al., 2023; Jiang et al., 2024; Zeng et al., 2024; Zhang et al., 2024b; Wang et al., 2024a) leverage monocular video as a condition to generate high-quality and naturally dynamic 4D objects. Nevertheless, generating 4D scenes remains challenging for these methods, as they often require monocular videos with clear foreground subjects, which are difficult to obtain. The text-to-4D method (Xu et al., 2024; Bahmani et al., 2024a), and the monocular-video-to-4D method (Chu et al., 2024), can generate 4D scenes, but they struggle with situations involving geometrical 4D scene transitions. To address this, we propose TRANS4D, which enables the stable and convenient generation of 4D scenes with physical 4D transitions.

### 2.2 GENERATION WITH LARGE LANGUAGE MODEL

Inspired by the advancements in LLMs and MLLMs (Touvron et al., 2023; Liu et al., 2024a; Lin et al., 2023a; Hong et al., 2023; Qi et al., 2024), many works have leveraged these models to achieve higher-quality generation. In image generation (Dong et al., 2023; Yang et al., 2024a; Hu et al., 2024; Han et al., 2024; Berman & Peysakhovich, 2024) and image editing (Fu et al., 2024; Li et al., 2024; Jin et al., 2024; Tian et al., 2024a; Yang et al., 2024b), LLMs are first utilized to enhance the quality of output images. Thanks to the powerful planning abilities of LLMs, these image generation and editing methods can handle more complex scenarios. Subsequently, with the research surge sparked by Sora (Liu et al., 2024b), more and more video generation methods (Bao et al., 2024; Wu et al., 2024c; Tian et al., 2024c; Maaz et al., 2024) and storytelling approaches (Soldan et al., 2021; Tian et al., 2024b; Yang et al., 2024c) have harnessed the impressive capabilities of LLMs to achieve coherent and realistic video synthesis, significantly contributing to the multimedia industry's development. Furthermore, with advancements in text-to-3D techniques (Poole et al., 2023; Lin et al., 2023b; Wang et al., 2024b; Zeng et al., 2023; Liang et al., 2024), some 3D (Sun et al., 2023; Feng et al., 2023; Chen et al., 2024b; Zhou et al., 2024) and even 4D (Xu et al., 2024; Wang et al., 2024a; Chu et al., 2024) generation methods now involve LLMs to produce high-fidelity 3D or 4D outputs with complex geometrical structures based on simple conditions. However, simultaneously planning temporal progression and spatial layout remains challenging for existing LLM and MLLM methods, making generating highly complex 4D scenes difficult. In this work, we equip MLLMs with enhanced capabilities for 4D planning, enabling more effective generation of complex 4D scenes.

### 2.3 TRANSITION GENERATION

According to the current research landscape, video transition synthesis is less explored than the more popular text-to-video and image-to-video generation methods. However, this direction is crucial in generating complex scenes and long stories. Scene transitions link two consecutive periods smoothly through location, setting, or camera viewpoint changes. This seamless transition ensures the coherent progression of the scene or story. Before video scene transitions, related research primarily focused on non-deep learning algorithms with fixed patterns, as well as Morphing (Wolberg, 1998; Shechtman et al., 2010) that identify pixel-level similarities and generative models (Van Den Oord et al., 2017; Gal et al., 2022) that leverage latent features of linear networks to achieve smooth and reliable transitions. Recent works (Chen et al., 2023; Ouyang et al., 2024; Xing et al., 2024; Feng et al., 2024; Zhang et al., 2024a) have advanced the field by enabling smooth and creative video transitions, paving the way for the creation of story-level, long-form videos. In addition to the video transition, our work first involves the geometry-aware transition into the text-to-4D synthesis.

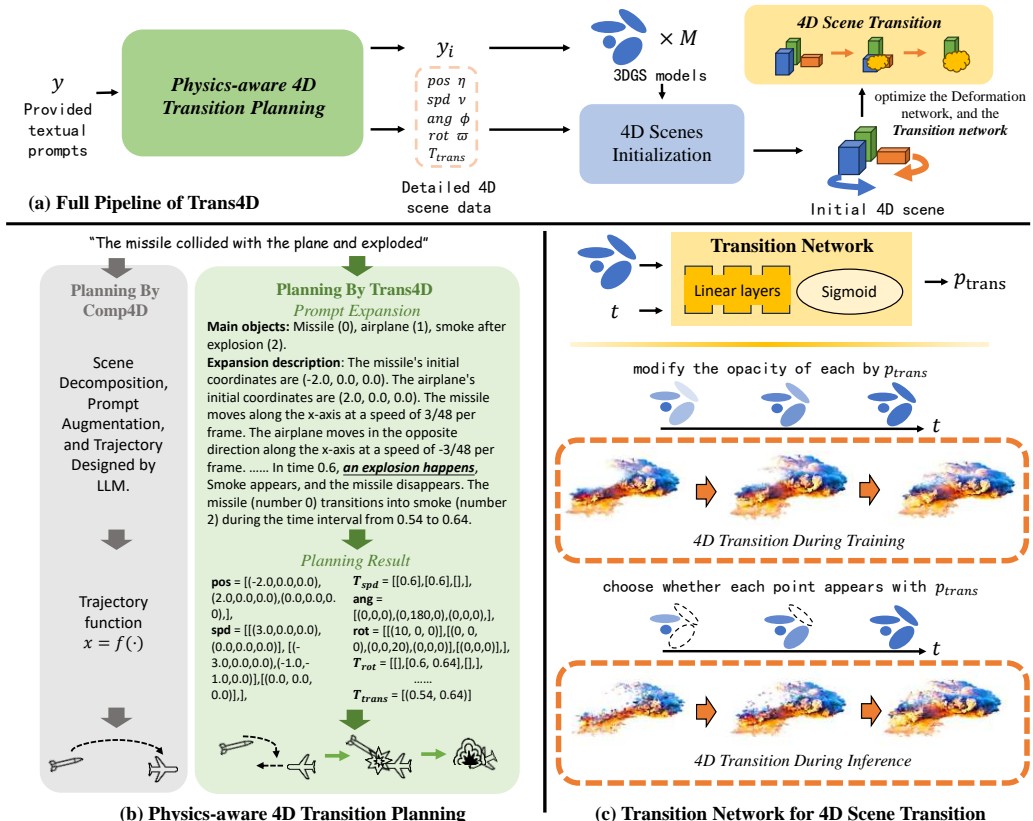

Figure 2: Overview of our TRANS4D, consisting of physics-aware 4D Transition Planning and Transition Network that enable 4D scene generation with complex interaction.

## 3 TRANS4D

Our TRANS4D is designed to achieve reasonable physical 4D scene transitions, as illustrated in Fig. 2(a). This section will explain how TRANS4D performs physics-aware 4D scene planning and accomplishes geometry-aware 4D transitions.

### 3.1 PRELIMINARIES

**3D Gaussian Splatting.** 3D Gaussian Splatting (3DGS) (Kerbl et al., 2023) $G$ consists of $N$ Gaussian points $\{g_i, i = 1, 2, ..., N\}$, and each point is defined with a center position $\mu$, covariance $\Sigma$, opacity $\alpha$, and color $c$. Each point $g_i$ is represented by a Gaussian distribution, and during rendering, the formula can be expressed as:

$$G(x) = \sum_{i=1}^{N} \alpha_i \cdot c_i \cdot \exp\left(-\frac{1}{2}(x - \mu_i)^\top \Sigma_i^{-1}(x - \mu_i)\right), \tag{1}$$

where $x$ is an arbitrary position in space during the rendering process.

**Text-to-4D Generation.** Before introducing our method, defining the input and output of the text-to-4D scene generation task is essential. In this work, the input is a text prompt $y$, and the output is a 4D scene represented by $M$ 3D Gaussian Splatting (3DGS) models $\{G_i, i = 1, 2, ..., M\}$, along with a deformation network $\{D_i, i = 1, 2, ..., M\}$ corresponding to each 3DGS model. Typically, the deformation network is represented by a multi-layer perceptron (MLP):

$$D(x, q, t) = (\Delta x_t, \Delta q_t), \quad x_t = x + \Delta x_t, \quad q_t = q + \Delta q_t, \tag{2}$$

where $x$ and $q$ denote the arbitrary position and orientation within the 3DGS model, and $x_t$ and $q_t$ represent the corresponding position and orientation at time $t$.

## 3.2 PHYSICS-AWARE 4D TRANSITION PLANNING

To optimize the 4D scene effectively, it is crucial to plan the placement and trajectories of objects within the generated scene based on the textual prompts $y$. This process includes determining which objects to generate, as well as specifying their initial positions $\eta$, movement speeds $v$, initial orientation angles $\phi$, rotational speeds $\varpi$, and scene transition times $T_{trans}$. Unlike Comp4D (Xu et al., 2024), which only uses LLMs to predict simple trajectory functions for 4D synthesis, our TRANS4D method leverages MLLM vision-language priors and introduces a physics-aware prompt expansion and transition planning approach. This advancement facilitates more reliable and complex initialization of 4D scenes.

**Physics-aware Prompt Expansion and Transition Planning.** The target of 4D planning is to derive spatial and temporal information from a given textual prompt. However, spatiotemporal data in a 4D scene are abstract and complex, making it difficult for LLMs or MLLMs to directly interpret and generate accurate physics-aware 4D scene data from a simple textual prompt. To overcome this challenge, we propose a physics-aware 4D prompt expansion and transition planning method. First, the method applies physical principles to analyze the original prompt, deriving spatiotemporal information and decomposing it into scene prompts $\{y_i, i = 1, 2, ..., M\}$. These prompts guide the creation of 3D objects within the scene. By utilizing both these prompts and the language-vision priors of MLLM, we extend the original textual input into a comprehensive, physics-aware scene description for the target 4D scene. This description provides specific details, including the placement of objects, their movements, and rotations along the x, y, and z axes over time, as well as key events (e.g., changes in motion speed or the appearance and disappearance of objects). By converting this description into a specific data format, the desired 4D scene data is obtained. As illustrated in Fig 2(b), this method enables MLLM to generate detailed and physically plausible 4D scene data, including $\eta$, $v$, $\phi$, $\varpi$, and $T_{trans}$. The detailed reasoning prompts are provided in the Appendix.

**Initialization of 4D Scene.** Based on the $\{y_i, i = 1, 2, ..., M\}$ obtained through the planning method, we utilize SDS with text-to-image generation model (Ye et al., 2023) to guide basic 3DGS models $\{G_i, i = 1, 2, ..., M\}$ synthesis. Using the planning 4D scene data, We calculate the transformation function for any position within these 3DGS models at each time $t$ as:

$$x = R(\phi + \varpi \cdot t)x_\xi + \eta + v \cdot t \qquad (3)$$

where $R$ denotes the rotation matrix, $x$ represents the arbitrary position in the 3DGS model, and $x_\xi$ is the coordinate of $x$ when the 3DGS model is at $(0, 0, 0)$. By integrating $\{G_i, i = 1, 2, ..., M\}$ with the transformation function, we obtain an initial 4D scene.

After obtaining the physics-aware planning, we use geometry-aware 4D transitions to effectively visualize the physical dynamics derived from this planning. In the next section, we detail how our proposed transition network realizes these geometry-aware 4D transitions.

## 3.3 GEOMETRY-AWARE 4D TRANSITION

By utilizing the initial 4D scene and the deformation network, we can achieve 4D scene synthesis in certain scenarios through global object positioning and local dynamics. However, depending exclusively on movement is limited, as it cannot support geometry-aware 4D transitions that involve significant object deformation, such as the appearance or disappearance of objects in a 4D scene.

To overcome this limitation, we propose a geometry-aware Transition Network (TransNet), which is a multi-layer perceptron (MLP) with a Sigmoid activation function at the output layer. As shown in Fig. 2(c), TransNet takes the position of the point cloud and the time $t$ as inputs, and processes them through several linear layers to produce an intermediate output. This intermediate output is then scaled by a coefficient $w_{trans}$ before inputting into the final Sigmoid function. The final output of TransNet denotes as $p_{trans}$, which lies between 0 and 1 and serves as a reference for 4D transition.

$$p_{trans} = \sigma(w_{trans} \cdot h(x_t, q_t, t)), \qquad (4)$$

where $h(x_t, q_t, t)$ represents the intermediate output from the linear layers of TransNet, $\sigma$ is the Sigmoid activation function, and $w_{trans}$ is a scaling coefficient, typically set to 10 or higher, to amplify the changes of the point cloud over time $t$.

During the training stage, to ensure that TransNet is differentiable, we modify the opacity of each point cloud by multiplying the opacity $\alpha$ directly with $p_{trans}$. During the inference stage, to ensure a noticeable transition, $p_{trans}$ is used to determine whether each Gaussian point of the 3DGS model appears in the 4D scene. This method enables a smooth and natural 4D scene transition. The calculation process is as follows:

$$B = \begin{cases} 1, & \text{with probability } p_{trans}, \\ 0, & \text{with probability } 1 - p_{trans}, \end{cases} \tag{5}$$

When $B = 1$, the point cloud appears in the 4D scene; otherwise, it does not. Compared to manually constraining the number of points in the 3DGS model at different time intervals, TransNet allows for flexible and rational control of point variations during the transition process, effectively achieving desired geometry-aware 4D scene transitions.

### 3.4 Efficient 4D Training and Refinement

Conventional text-to-4D optimization strategies typically rely on SDS loss based on video DM to produce 4D results with reliable dynamics, which incurs high computational costs. To efficiently achieve high-fidelity 4D scene synthesis with realistic dynamics, we optimize TRANS4D in two phases: first, we train the deformation network and TransNet using 3DGS models with a relatively small number of point clouds, minimizing costs even with SDS based on video DM. Then, we refine 3DGS models, allowing for increased point cloud counts with lower computational overhead.

During the training of the deformation network and TransNet, the number of points in each 3DGS model is fixed at 20,000. We represent the rendered images of the 4D scene over 16 consecutive times $t$ as $\{\mathcal{I}^1, \mathcal{I}^2, ..., \mathcal{I}^{16}\}$. For SDS loss, noise is added to the rendered images, represented as $\mathcal{I}_t^1, \mathcal{I}_t^2, ..., \mathcal{I}_t^{16}$ at timestep $t'$. We optimize deformation network and TransNet using SDS based on video DM $\epsilon_{vid}$, which can be expressed as:

$$\nabla_{\theta_{dyn}} \mathcal{L}_{SDS-vid}(\{\mathcal{I}^1, \mathcal{I}^2, ..., \mathcal{I}^{16}\}, y) =$$
$$\mathbb{E}_{t', \epsilon} \left[ w(t') \left( \epsilon_{vid}(\{\mathcal{I}_{t'}^1, \mathcal{I}_{t'}^2, ..., \mathcal{I}_{t'}^{16}\}, y, t') - \epsilon \right) \frac{\partial \{\mathcal{I}^1, \mathcal{I}^2, ..., \mathcal{I}^{16}\}}{\partial \theta_{dyn}} \right], \tag{6}$$

where $\theta_{dyn}$ represents the parameters of deformation network and TransNet. During this training stage, the points in the 3DGS model are neither cloned nor split, ensuring efficient training of both networks. To further enhance the quality of the 4D scenes, we use an SDS loss based on text-to-image DM $\epsilon_{img}$ to supervise further optimization of the 3DGS model. At this stage, the points in the 3DGS model are cloned and split for the refinement:

$$\nabla_{\theta_G} \mathcal{L}_{SDS}(\mathcal{I}, y) = \mathbb{E}_{t', \epsilon} \left[ w(t') \left( \epsilon_{img}(\mathcal{I}, y, t') - \epsilon \right) \frac{\partial \mathcal{I}}{\partial \theta_G} \right], \tag{7}$$

where $\mathcal{I}$ represents the rendered result of the 4D scene at a random time $t$, and $\theta_G$ represents the parameters of the 3DGS model. Meanwhile, the inputs to the deformation network and TransNet consist solely of the positions of the 3DGS model's points. Therefore, even after the refinement stage, while the 3DGS models in the 4D scene become more detailed and realistic, the dynamics of the 4D scene remain unaffected.

## 4 Experiments

**Implementation Details.** In this work, all experiments are conducted on four A100-SXM4-80GB GPUs. In Stage 1, we optimize for 5000 steps using the Adam optimizer (Kingma, 2014) to obtain the 3DGS models. In Stage 2, we perform 4500 optimization steps to train the deformation network and the Transition Network. During the refinement phase, we further optimize the 3DGS models for objects that cannot be represented in high quality with only 20000 points (e.g., complex structures like "volcano"). This refinement is performed over 4000 steps using the SDS loss. We ensure a fair comparison by using the same models across all methods for both generation and supervision. For any use of a text-to-image generation model, we use Stable Diffusion 2.1 (Rombach et al., 2022); for any use of a multiview generation model, we use MVDream (Shi et al., 2024); and for any use of a text-to-video generation model, we use VideoCraft (Chen et al., 2024a).

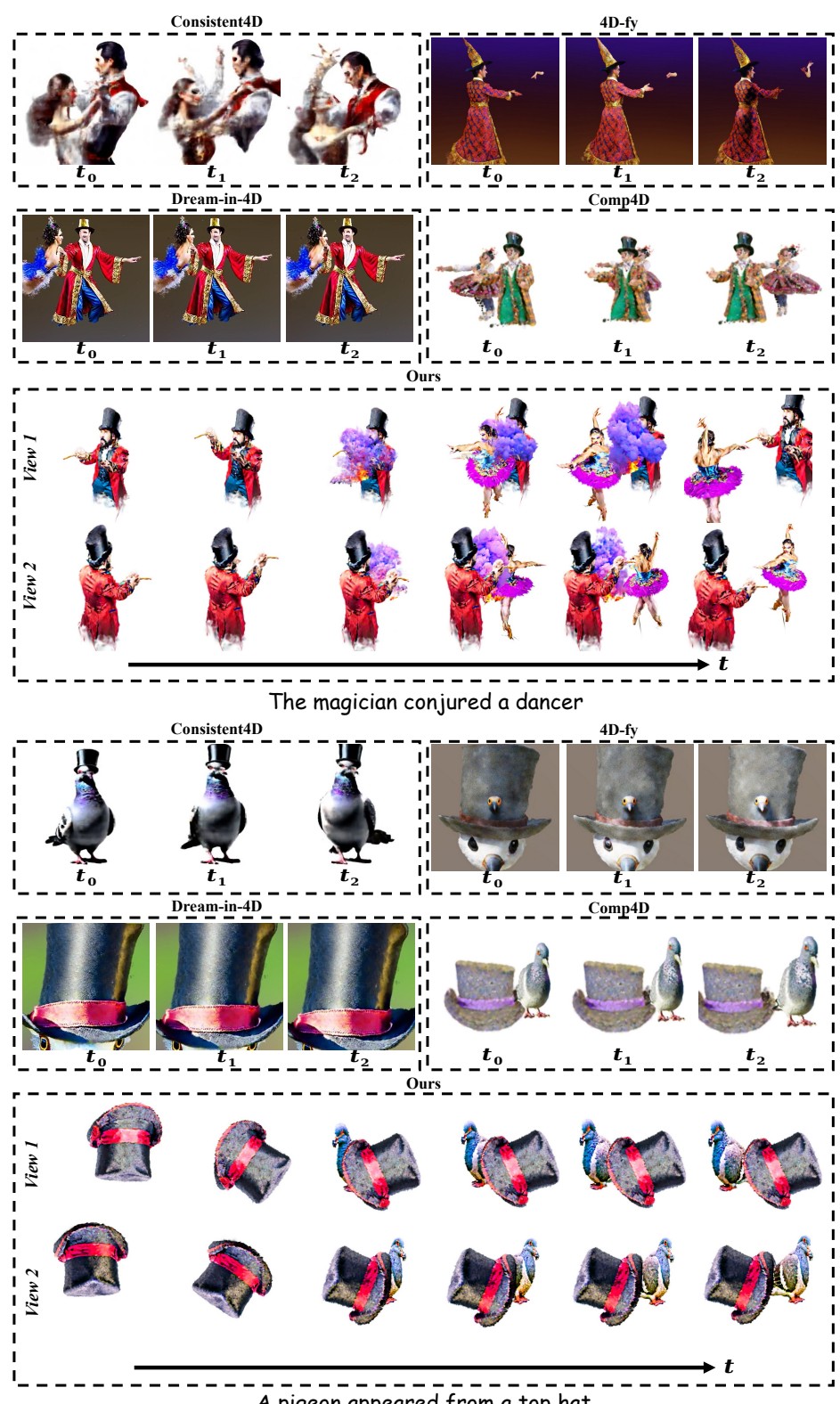

Figure 3: Qualitative comparison with previous baseline methods (Bahmani et al., 2024b; Zheng et al., 2024; Jiang et al., 2024; Xu et al., 2024). Our method achieves smoother geometric 4D transitions and produces more realistic object interactions within 4D scenes.

Table 1: Quantitive comparison of text-to-4D generation.

| Metrics | Consistent4D | 4D-fy | Dream-in-4D | Comp4D | TRANS4D (Ours) |
|---|---|---|---|---|---|
| QAlign-vid-quality ↑ | 2.275 | 3.017 | 3.035 | 2.961 | **3.226** |
| QAlign-vid-aesthetic ↑ | 1.924 | 2.089 | 2.111 | 1.774 | **2.148** |
| Vid-MLLM-metrics ↑ | 0.5931 | 0.4347 | 0.5063 | 0.5532 | **0.6483** |
| CLIP-score ↑ | 0.2836 | 0.2661 | 0.2607 | 0.2757 | **0.2941** |
| User study ↑ | 0.72 | 0.64 | 0.67 | 0.59 | **0.78** |

**Baseline Methods.** To validate the effectiveness of our method in generating complex 4D scenes with geometry-aware 4D transitions, we compare it with several different 4D generation methods. These methods include text-to-4D-object methods 4D-fy (Bahmani et al., 2024b) and Dream-in-4D (Zheng et al., 2024), a monocular-video-to-4D-object method Consistent4D (Jiang et al., 2024), and a text-to-4D-scene method Comp4D (Xu et al., 2024).

**Metrics.** Due to the lack of visual ground truth in text-to-4D generation tasks, we employ QAlign-vid-quality and QAlign-vid-aesthetic metrics (Wu et al., 2024b) to evaluate the quality and aesthetics of the generated 4D scenes. To assess the semantic alignment of the generated results, we utilize the CLIP-score (Park et al., 2021) and MLLM-score. Additionally, we conduct a user study to enhance the credibility of our comparison results. More details on QAlign-vid-quality, QAlign-vid-aesthetic, CLIP-score, MLLM-score, and the user study are provided in the Appendix.

### 4.1 TEXT-TO-4D SYNTHESIS

**Quantitative Results.** To assess the effectiveness of TRANS4D in complex 4D scene synthesis, we utilize 30 complex textual prompts for 4D scene synthesis. Most of these prompts involve geometry-aware transitions, with the specific prompts detailed in the supplementary material. As shown in Table 1, TRANS4D surpasses other methods across all metrics. The text-to-4D methods, 4D-fy and Dream-in-4D, achieve high scores on the metrics utilized Q-align, demonstrating their ability to generate high-quality 4D scenes. However, they perform poorly on the CLIP and MLLM scores, highlighting that it remains challenging for them to generate 4D scenes that accurately align with the input text. Additionally, our TRANS4D achieved the highest score in the user study, further validating its effectiveness.

**Qualitative Results.** To intuitively demonstrate the superiority of our method in generating complex 4D scenes, that have significant object deformations, we conduct a qualitative comparison with other baseline models. As shown in Fig. 3, the rendered videos of the 4D outputs generated by our method exhibit the most reasonable and high quality. Additionally, while 4D-fy and Dream-in-4D also produce high-quality visual outputs, these text-to-4D-object generation methods struggle to create 4D scenes with coherent dynamics based on textual requirements. Lastly, the results from Consistent4D indicate that monocular-video-to-4D generation methods perform better for simple 4D object generation. However, when the monocular video involves complex dynamics and interactions (as in the visualization example, "The magician conjured a dancer"), these methods struggle to produce satisfactory 4D outputs. Moreover, acquiring a monocular video with both clear subjects and reasonable dynamics is inherently challenging. Therefore, our TRANS4D is currently the most convenient and reliable method for generating complex 4D scenes.

### 4.2 MODEL ANALYSIS

To highlight the key contributions of TRANS4D, including Physics-aware 4D Transition Planning and the Transition Network, we conduct additional user studies to demonstrate the effectiveness of our proposed models. Furthermore, we incorporate visual comparisons to showcase the necessity and benefits of refinement.

**Rationality of MLLM-planned Trajectory.** We have demonstrated that our method for initializing 4D scenes outperforms the simple function-based method Comp4D. To further showcase the advantages of our Physics-aware 4D Transition Planning method, we conduct an experiment

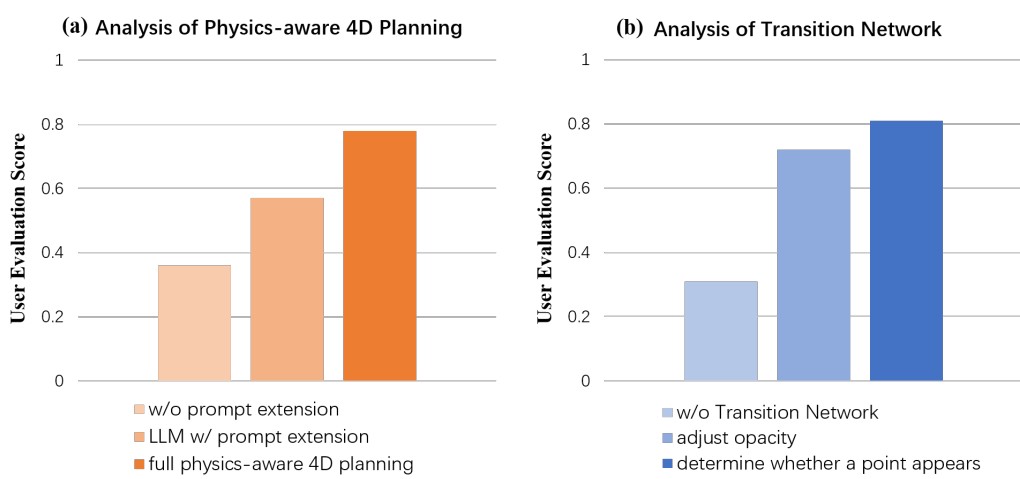

Figure 4: Additional user study for model analysis.

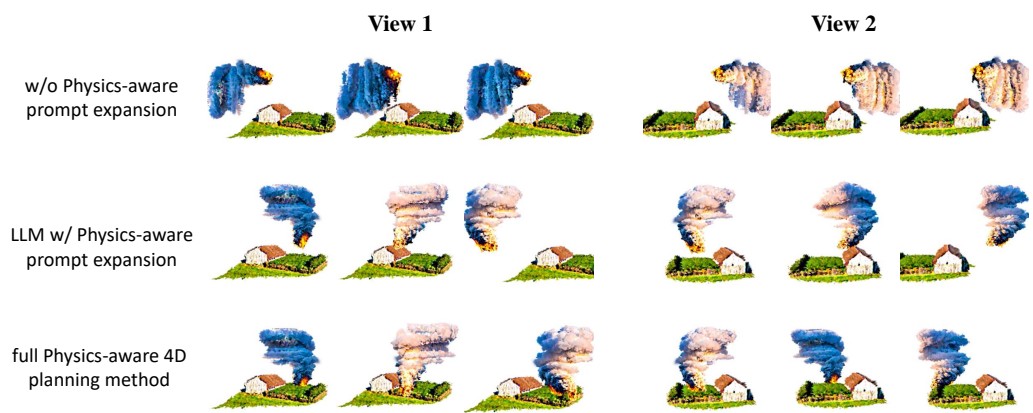

Figure 5: Ablation study of Physics-aware 4D Transition Planning method.

where volunteers evaluate videos generated from three different initialization methods: (1) Without Physics-aware prompt expansion: the MLLM receives only one example (including input text and 4D data) to generate 4D scenes based on other input texts; (2) Utilizing an LLM to predict the 4D data with Physics-aware prompt expansion; and (3) Our complete Physics-aware 4D Transition Planning method. As shown in Fig. 4(a), without Physics-aware prompt expansion, the MLLM struggles to generate plausible 4D data for scene initialization, resulting in poor outcomes. This underscores the importance of physics-aware prompt expansion. Moreover, when we utilize the LLM to produce the 4D data with Physics-aware prompt expansion, the predicted 4D data lack precision due to the absence of vision-language priors. As illustrated in Fig. 5, incorporating the full Physics-aware 4D Transition Planning method significantly enhances the results, highlighting its ability to enrich our approach with prior knowledge for more reasonable scene initialization.

**Geometrical Expressiveness.** To better observe the effects of the transition network, we decelerate the geometric-aware 4D transition process, allowing volunteers to discern the transition effects. We provide the volunteers with three different videos representing various transition methods: (1) without using the transition network; (2) using the transition network, where $p_{trans}$ is multiplied by the opacity; and (3) using the transition network, where $p_{trans}$ determines which points should appear. The volunteers are asked to evaluate which process appears more natural. As shown in Fig. 4(b), it is evident that the majority of volunteers find the transitions incorporating the transition network to be more natural, with the point selection method receiving the highest scores due to the clearer and more distinct transition. We demonstrate the generated results in Fig. 6, which highlights the pivotal significance of the proposed transition network in this study.

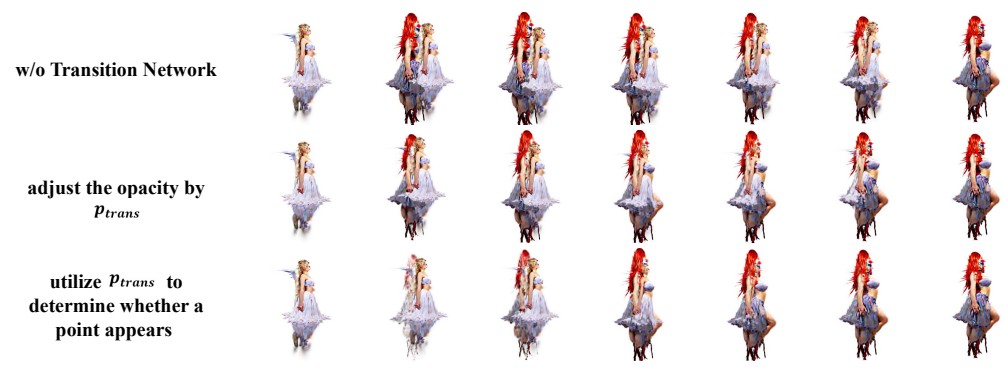

Figure 6: Ablation study of Transition Network.

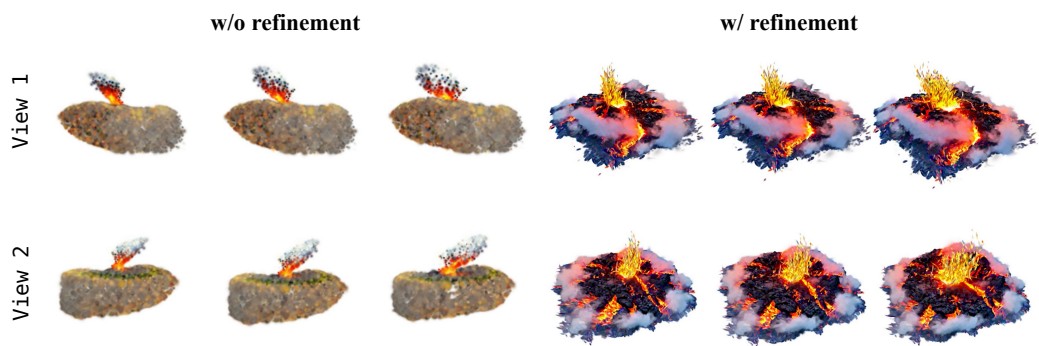

Figure 7: Ablation study of refinement.

**Efficiency and Quality of Refinement.** When a 4D scene contains over 200,000 point clouds, directly supervising it with video SDS loss consumes **80GB or more** GPU memory limit, while leading to suboptimal quality. In contrast, by separating the training process, we reduce memory usage to around **50GB**, almost halving the requirement, while significantly improving the quality of the generated 4D scene. Specifically, we initially represent the 4D scene using minimal point clouds while training the deformation and transition networks. Then, we apply a refinement process to improve the quality of each 3DGS model by increasing the number of point clouds as needed. This stepwise training manages memory efficiently while producing high-quality 4D scenes. As demonstrated in Fig. 7, for massive 3D objects like "volcano erupting", sparse point clouds cannot represent them effectively. Hence, refining such 3D objects is essential. In conclusion, our training strategy balances efficiency and quality, enabling the generation of high-quality 4D scenes with relatively limited computational resources.

## 5 CONCLUSION AND FUTURE WORK

In this work, we propose TRANS4D, a novel text-to-4D scene generation method that produces high-quality 4D scenes involving complex object interactions and significant deformations. Specifically, we introduce a Physics-aware 4D Transition Planning method, which enables MLLM to initialize realistic 4D scenes with multiple interacting objects. To facilitate geometry-aware transitions in the generated 4D scene, we design a Transition Network that dynamically determines whether each point cloud in the 4D scene should appear or disappear, allowing our method to handle substantial object deformations naturally. Our experiments demonstrate that TRANS4D consistently generates high-quality 4D scenes with complex interactions and smooth, geometry-aware transitions.

For future work, We will continue to improve the quality of multi-object interactions in 4D scenes, which will help achieve more realistic 4D scene generation, and support the development of the video multimedia and gaming industries.

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

# A   APPENDIX

In Appendix A.1, we provide detailed information of our evaluation metrics. Appendix A.2 outlines the specific prompts and process of the Physics-aware 4D Transition Planning method. Finally, in Appendix A.3, we present the textual prompts used for evaluation along with additional 4D scenes generated by TRANS4D.

Table 2: 4D scene decomposition.

```
You are a 4D Scene Decomposing Agent

Your task is to decompose the 4D scene into several
    appropriate parts based on the prompt provided by the
    user. Unlike 3D scene generation methods that only need to
    split according to the content of the prompt, you need to
    analyze the possible physical dynamic that may occur in
    the provided prompt from both temporal and spatial
    dimensions. Concurrently, based on the analysis results,
    decompose the provided prompt into several prompts in the
    time-space dimension.

--- Main Object ---

These objects' prompts will be used for generating 3D objects
    first, and then add time dimension to generate a complete
    4D scene.
Therefore, if the same object undergoes significant physical
    changes over time, it should be considered as two separate
    main objects.
The scene's background is blank, and only moving objects,
    suddenly appearing objects like clouds and smoke, and
    objects undergoing shape changes, such as melting or
    breaking, need to be considered.

--- Examples ---
......
```

## A.1   DETAILS OF METRICS

In this section, we provide a more detailed explanation of the metrics and user studies discussed in the main paper.

**QAlign-vid-quality and QAlign-vid-aesthetic.**   Q-Align (Wu et al., 2024b) is a large multi-modal model fine-tuned from mPLUG-Owl2 (Ye et al., 2024) using extensive image and video quality-assessment datasets. It has demonstrated strong alignment with human judgment on existing quality assessment benchmarks. In line with Comp4D (Xu et al., 2024), we use Q-Align to evaluate the quality of the generated 4D scenes. Specifically, we input rendering videos of 4D scenes produced by various methods from viewpoints of -120°, -60°, 0°, 60°, 120°, and 180° into Q-Align. The output scores from Q-Align range from 1 (worst) to 5 (best). We calculate the average score of these outputs to compare the performance of different 4D generation methods quantitatively.

**CLIP score.**   The CLIP score (Park et al., 2021) is a widely used metric for evaluating the correlation between input textual prompts and generated images. Following the approach in 4D-fy (Bahmani et al., 2024b), we calculate the CLIP score between the frames of the rendered videos and the input textual prompts. Due to the complexity of 4D scene generation, which involves significant object dynamics, we use the maximum CLIP score obtained across all frames of each rendered video as the

Table 3: Complete Scene Expansion Description

```
You are an Efficient Scene Expansion Agent.

Your task is to use these decompositional main objects and the
    prompt to expand the provided prompt into a complete
    physics-aware 4D scene description.

--- Scene ---

The scene is a 4D video clip composed of the main objects
    extracted earlier. The scene information should include:

- The initial position of each object, represented in the form
    [x, y, z].
- The movement path of the objects defines the movement vector
    per frame. Each object can have multiple movement segments.
- The time points when movements start or stop.
- The initial rotation angle of the objects is expressed in
    degrees as [rx, ry, rz] (rotation along the x, y, and z
    axes respectively).
- The rotation path of the objects, defining the rotation
    change per frame.
- The time intervals when rotations occur.
- The time states of the objects, such as when they appear,
    disappear, or transform at specific times.
- The transformation relationships between objects, specifying
    which objects transform into each other during certain
    time intervals and when these transformations occur.

The time points are represented within a single 4D segment,
    with 0 indicating the start and 1 indicating the end.
    Other states use decimals to specify the exact time point
    within the segment.

The scene's center is [0, 0, 0], and the range for each
    coordinate axis within the scene is [-1, 1]. Positions
    outside this range are considered outside the scene.
    Objects can enter the scene from outside, but each main
    object must appear within the scene at some point.

--- Examples ---
......
```

representative score. To evaluate their performance, we compare the average CLIP scores of rendered videos generated by different methods.

**MLLM score.** Although the CLIP score is a commonly used metric to evaluate semantic alignment, it can not fully analyze the reasonability of rendered videos. To more effectively evaluate the semantic alignment of the generated 4D results, we propose the MLLM score which leverages the vision-language knowledge of GPT4o to evaluate the correlation between the rendered videos and the input textual prompts. Specifically, we present the rendered videos and the provided textual prompts for the ChatGPT-4o. The specific prompt provided for ChatGPT-4o scoring the semantic alignment as: "We provide several <video> clips along with a <text prompt>. The videos represent rendered 4D scenes from specific viewpoints. Please evaluate the 4D scenes generated by different methods based on the alignment between the video and the text prompt, as well as the overall video quality, and assign a score between 0 and 1."

Table 4: The specific prompt for obtaining 4D planning data.

```
You are a 4D data production Agent.

Your task is to transfer the complete 4D scene description
    into precise 4D planning data.

The output should be in the json format:
{
    "sample": {
        "obj_prompt": [
            "List of objects involved in the scenario"],
        "TrajParams": {
            "init_pos": [
                [x, y, z] // Initial positions of objects in 3D
                    space],
            "move_list": [
                [
                    [dx, dy, dz], // Movement vector
                    [dx, dy, dz] // Additional movement after an
                        event
                ] ],
            "move_time": [
                [time] // List of times when movements occur or
                    stop],
            "init_angle": [
                [rx, ry, rz] // Initial rotation angles (degrees)
                    of objects along x, y, z axes],
            "rotations": [
                [
                    [rx, ry, rz], // Rotation vector per frame
                    [rx, ry, rz] // Optional: Additional rotation
                        after an event
                ] ],
            "rotations_time": [
                [start_time, end_time] // Times when rotations
                    occur],
            ......
            "trans_list": [
                [obj_index, transition_obj_index] // Objects that
                    transition into each other],
            "trans_period": [
                [start_time, end_time] // The time period when the
                    transition occurs.]
        }
    }
}
```

**User study.** For unsupervised text-to-4D-scene generation, the user study is the most convincing metric. To further validate the effectiveness of our method, we conduct a comprehensive user study involving 80 volunteers. Each volunteer is randomly provided 10 test examples from the testing dataset introduced in this work. For each example, volunteers are asked to judge whether the generated results from various 4D methods successfully achieve the desired 4D synthesis based on the given text inputs. Volunteers rate each result on a scale from 0 to 1, where a score closer to 1 indicates better alignment with the expected outcome.

Table 5: Textual prompts used in the user study.

| |
|---|
| The missile collided with the plane and exploded. |
| A cavalry charged two shield-bearing infantry. |
| The magician conjured a dancer. |
| The ice block melts into water. |
| The volcano erupted violently. |
| The tree fell after being cut by the harvester. |
| The water balloon burst on impact. |
| The clock struck midnight. |
| The egg cracked open. |
| The spaceship took off from Earth and entered space. |
| The tornado formed over the plains. |
| The butterfly emerged from the cocoon. |
| The snowflake melted on the tongue. |
| The fish jumped out of the water. |
| The corn kernels pop into popcorn. |
| The moon appeared from behind the clouds. |
| A pigeon appeared from a top hat. |
| An angelic girl is becoming a puppet of the devil. |
| An explosion occurs while a wizard is brewing a magic potion. |
| A sage caused a gigantic flower to bloom. |
| Three worshippers pray for the appearance of an angel. |
| A zombie crawls out of the tombstone. |
| A dragon breathes fire onto a knight's shield. |
| A giant cracks the ground with its heavy footsteps. |
| A knight draws a glowing sword from a stone. |
| A sorcerer opens a portal to another dimension. |
| A ghost passes through a wall, leaving behind a cold mist. |
| A castle tower collapses after being struck by lightning. |
| A violin plays itself, filling the air with haunting melodies. |
| The appearance of the sun clears the fog. |

## A.2 MORE DETAILS OF OUR MODEL

**Physics-aware 4D planning.** Multimodal Large Language Models (MLLMs), leveraging their vision-language priors, have the potential to generate reasonable and natural spatiotemporal data. In this work, we leverage the spatiotemporal awareness of MLLMs to achieve impressive 4D scene initialization and 4D transition planning. During the process of obtaining 4D data, we require the MLLM to ensure that the generated plans are consistent with physical principles and geometrically coherent, thereby guaranteeing both physical plausibility and the correctness of spatial relationships. Specifically, Tables 2, 3, and 4, present the detailed prompts for scene decomposition, physics-aware 4D prompt expansion, and 4D planning data, respectively.

## A.3 ADDITIONAL RESULTS

**Textual prompts used for comparison.** In Table. 5, We provide the specific textual prompts used in quantitative comparison.

**comparison results.** In Fig. 8, we provide more generated 4D scenes of TRANS4D, to further demonstrate the effectiveness of our method.

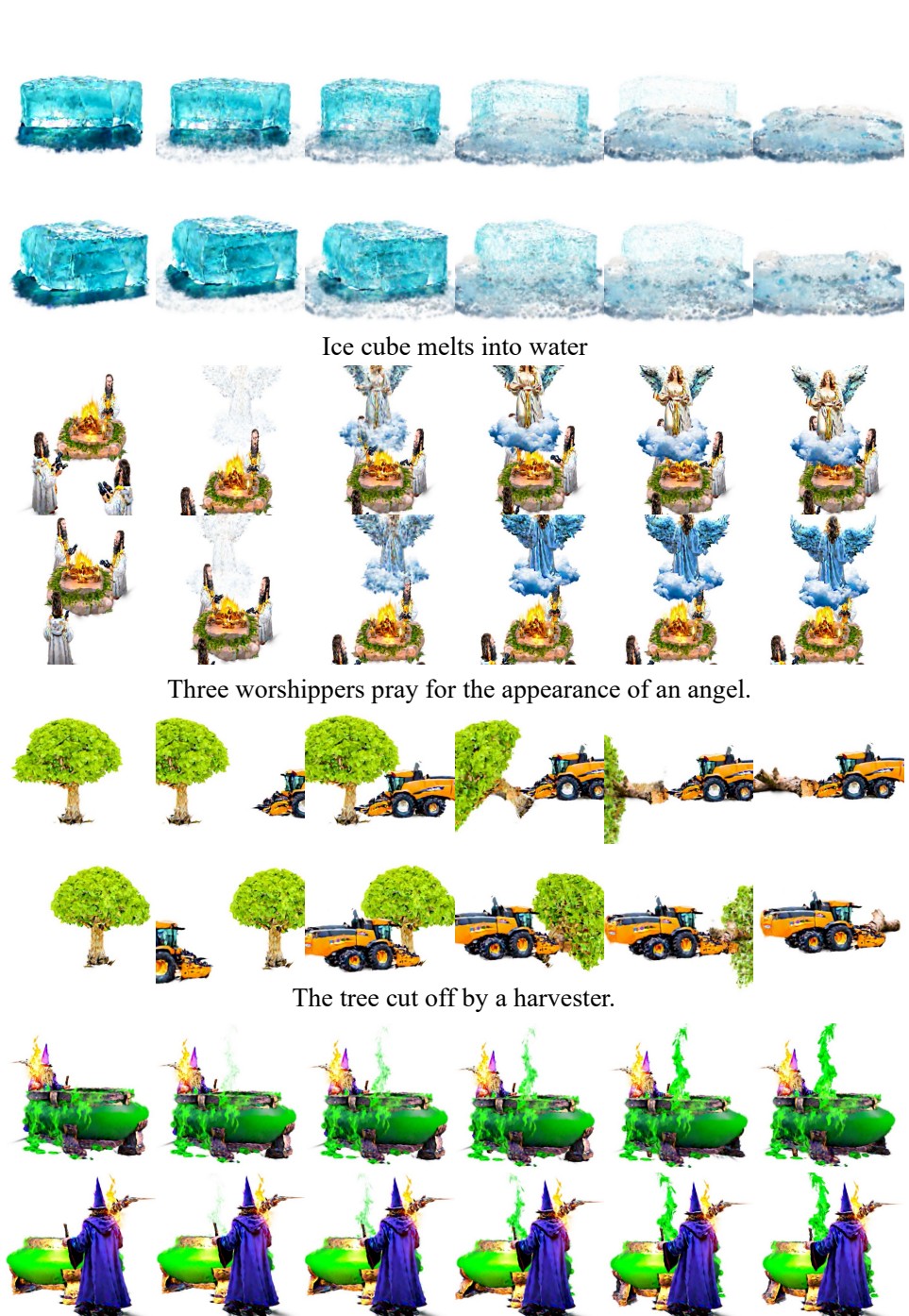

Ice cube melts into water

Three worshippers pray for the appearance of an angel.

The tree cut off by a harvester.

Green flames ignited during the wizard's process of brewing the potion.

Figure 8: Additional generated 4D results, our TRANS4D can consistently produce high-quality 4D scenes.

