# OpenReview forum: "Trans4D: Realistic Geometry-Aware Transition for Compositional Text-to-4D Synthesis"
_ICLR.cc/2025/Conference — ICLR 2025 Conference Withdrawn Submission_

### Official Review · Reviewer_1in4 · 2024-10-22

**Soundness:** 1
**Presentation:** 1
**Contribution:** 1
**Rating:** 3
**Confidence:** 5

**Summary:**

The paper proposes a framework to generate 4D videos of scenes consisting of multiple objects.

The paper proposes to first have the short video, the objects, and their approximate movements planned by a trained foundation model in the form of language.
Then the individual objects are synthesized over time, as well as their transitions using a transition network,

**Strengths:**

The paper proposes a reasonable problem that is interesting and it proposes a reasonable framework to tackle the problem in two steps. (However, only the high-level breakdown is reasonable).

**Weaknesses:**

The core concept of the transition network does not seem to be adequate to handle the complexity of the problem. The resulting examples showcase objects "jumping" into existence and not coherent and interesting videos involving multiple objects. I conjecture that a transition network alone is not sufficient to handle the complexities involved in object to object interaction.
Another reason may be that the model itself is not powerful enough or not trained long enough. The current interaction, that just inserts an object at a given time step is possible with the current framework, but this is not enough to tackle the problem.
The example of the magician and the dancer is adequate for this particular scene, as the dancer probably really should jumpt into existence. The example of the truck cutting down a tree shows the limitation of the method and does not really address the issues.

**Questions:**

The paper gives mixed information regarding the nature of the transition network. Some sentences indicate that an opacity is trained per Gaussian and other sentences indicate that an opacity is trained for each object. In either case, I can only observe results that seem to stem from a single opacity per object as I cannot perceive complex transitions. I would like to ask to get further clarification on this point in the rebuttal. The exploding plane does not seem to explode, rather it appears to rotate under a cloud of smoke. This is a failure case of the method. In addition, I cannot observe if there are any examples where the capability to create a smooth transition is used for an interesting effect. The paper reports that "TransNet allows for flexible and rational control of point variations during the transition process, effectively achieving desired geometry-aware 4D scene transitions." I am not able to observe that in any of the provided examples.

**Details Of Ethics Concerns:**

I would suggest changing the example of a missile shooting down a plane. This is a passenger plane, no less, carrying many passengers, and the paper uses the killing of maybe hundreds of people as the first prominent example in the paper. This is not really a reason to reject the paper or conduct an ethics review, but it's not something I would recommend.

---

### Official Review · Reviewer_r7Pp · 2024-10-31

**Soundness:** 3
**Presentation:** 1
**Contribution:** 3
**Rating:** 5
**Confidence:** 4

**Summary:**

The paper introduces TRANS4D, a framework designed to enhance 4D scene generation by incorporating realistic, geometry-aware transitions between objects and actions in virtual environments. Existing methods struggle with complex object deformation and scene interactions, particularly when generating 4D content from text prompts. TRANS4D addresses these limitations by employing a physics-aware transition planning mechanism, which uses multimodal language models to interpret scene descriptions that include physical properties and dynamic timing. Additionally, a novel Transition Network enables smooth, geometrically realistic transitions, such as a missile evolving into an explosion cloud, creating a natural flow in complex 4D scenes. Experimental results show that TRANS4D outperforms current approaches in terms of scene realism and alignment with textual prompts, verified through quantitative assessments and user studies. This advancement holds significant potential for industries in gaming and multimedia that demand high-quality, interactive 4D content, with future work aimed at refining multi-object dynamics and interactions.

**Strengths:**

1. TRANS4D’s geometry-aware transition network enables highly realistic transformations, significantly enhancing visual and spatial coherence in the generated 4D scenes.

2. By incorporating physics-aware transition planning, TRANS4D increases the realism of object movements and transformations, adding an authentic layer to simulated scenes.

3. The framework excels in converting textual prompts into intricate 4D scenes, making it highly suitable for applications in content creation, multimedia, and gaming that demand high-quality, text-driven 3D and 4D synthesis.

**Weaknesses:**

1. The model structure depicted in Fig.2 appears overly abstract, relying heavily on textual explanations. This results in a lack of clarity regarding the model's specific structural components. A more detailed schematic representation could improve comprehension.

2. Further clarification is recommended regarding the role of LLM in generating detailed and physically plausible 4D scene data. Elaboration on how the model ensures physical consistency in generated scenes would strengthen this section.

3. There is a lack of cost comparison with previous methods, and there is no explanation of how much is used as a result of quantitative experimental prompts.

4. Some key discussions are missing. Based on the prompts given for LLM outputs in this paper, it appears that simply providing a complete description enhances the results. In other words, the quality of 4D generation seems to rely heavily on initialization. If this is the case, the use of LLMs may not significantly impact the overall performance, thus reducing the novelty of this work.

**Questions:**

The quality of the generated visualizations is inconsistent: Fig. 6 and 7 are very clear, while Fig.1, 3, and 5 are quite blurry, making it hard to believe these results were produced by the same model.

---

### Official Review · Reviewer_eb7r · 2024-11-03

**Soundness:** 2
**Presentation:** 3
**Contribution:** 2
**Rating:** 3
**Confidence:** 4

**Summary:**

The paper targets at the task of generating 4D scenes with transitions. The paper utilizes the MLLMs to generate detailed 4D scene transition information, and adopts 3D Gaussians and transition networks to represent the 4D transition scenes. 30 complex textual prompts for 4D scene synthesis are utilized to evaluate the method.

**Strengths:**

++ Leveraging MLLMs for 4D scene transition generation planning is promising.

++ The coarse-to-fine training of 4D Gaussians is effective in improving efficiency.

++ Paper is well written and presented.

**Weaknesses:**

-- The scene transition network can only represent the appearing and disappearing of each Gaussian, and cannot represent complex scene transition effects such as appearance changing, interactions, etc. This heavily limits the scene transition types of the proposed method.

-- Generated results are blur and contain artifacts. For example, in the supplementary video of "magician_dancer", the dancing person contain three arms.

-- Unfair comparison. The compared methods are mainly text-to-4D generation approaches that do not have any scene transition related designs. It seems a bit unfair to compare those methods with the proposed method mostly on the text prompts with transitions, as described in L403-404. It would be better to also include comparisons on the non-transition text prompts to extensively demonstrate the superiority of the proposed method.

**Questions:**

How to set the scaling coefficient of Eq. (4)? Is this parameter fixed for different prompts?

---

### Official Review · Reviewer_cfQT · 2024-11-04

**Soundness:** 2
**Presentation:** 3
**Contribution:** 2
**Rating:** 5
**Confidence:** 5

**Summary:**

The paper introduces Trans4D for generating large deformation within the text-to-4D generative models. The authors divide the process into three steps. Firstly, they propose a physics-aware 4D transition planning based on multi-modal large language models, providing the base for 4D scene initialization. Secondly, Trans4D includes a geometry-aware 4D scene transition module. Specifically, this module will determine whether a Gaussian point will appear at each timestep. Finally, a refining process has also be used to further improve the result's quality. Extensive experiments have been conducted and provided to demonstrate the performance and effectiveness of the proposed network.

**Strengths:**

+ The paper tackles the interesting and important task of bringing more deformation to current text-to-4D generative pipelines.

+ The paper is well-written and easy to follow.

+ The proposed geometry-aware 4D transition planning and geometry-aware 4D scene transition module is reasonable and showcased to be useful.

**Weaknesses:**

- The quality is still far away from satisfaction. See results in Fig.1 and Fig.3. The results are not good when two different objects interact with each other. The same issues can also be found in the video provided in the supplementary. I recognize it as the most severe issue of this paper.

- The geometry-aware 4D transition network will only predict the appearance of each point, which may be the cause for the previous weakness. Meanwhile, it may also make this method not suitable for modeling articulate deformation, limiting the applications of the proposed Trans4D.

- More comparisons with image-to-4D or video-to-4D pipelines would be better.

**Questions:**

Please see the above strengths and weaknesses

---

### Note · Authors · 2024-11-15

I have read and agree with the venue's withdrawal policy on behalf of myself and my co-authors.